# The Importance of Social Influencer-Generated Contents for User Cognition and Emotional Attachment: An Information Relevance Perspective

Xiuping Zhang and Jaewon Choi *

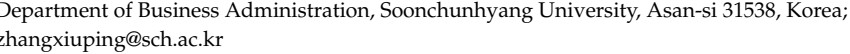

Department of Business Administration, Soonchunhyang University, Asan-si 31538, Korea;
zhangxiuping@sch.ac.kr
* Correspondence: jaewonchoi@sch.ac.kr

**Abstract:** It has become a marketing trend for marketers to use influencers to advertise and sell products because influencers can affect the attitudes and decision-making of other social media users. Most previous research on influencer marketing has concentrated on its effectiveness as a promotional tool. In contrast, there have been limited studies on the influencer-social media user relationship. The relationship that influencers have with other social media users is the foundation for the success of influencer marketing. Therefore, it is critical to investigate the factors that affect the influencers' relationships with other users. In accordance with the concept of information relevance, this study presents a model relating the various attributes of influencer-generated content with emotional attachment and information quality to examine the relationship between influencers and other social media users. The findings of a survey of 280 respondents indicate that the interestingness, novelty, reliability, and understandability of influencer-generated content can effectively increase users' emotional attachment to influencers. Reliability and understandability can also have a significant positive impact on information quality. This eventually inclines social media users to follow or recommend influencers to others, which can increase the popularity of influencers. This study helps researchers and marketers advance their understanding of influencers' relationships with other social media users and offers management-related recommendations for influencers and marketers.

**Keywords:** influencer marketing; information relevance; emotional attachment; social media influencer

## 1. Introduction

In the sphere of marketing, businesses are turning to collaborating with social media influencers to help them advertise their commodities and brand to consumers [1]. Influencers are individuals who constantly share valuable content on social media to attract followers, and they have the ability to impact the opinions and purchasing decisions of other users [2]. It has already been demonstrated that marketing initiatives including social media influencers provide better results than those involving traditional celebrities [3]. Therefore, most companies that work with influencers are satisfied with the marketing results, and 68% of them plan to expand their budget in collaborating with influencers in the coming year [4]. This leads to predictions that the global influencer marketing market will surpass USD 370 million by 2027 [5].

It is critical for marketers to collaborate with the most appropriate influencer in order to efficiently capitalize on influencer marketing and reap the maximum benefit from it [6]. The follower count that influencers have is a crucial factor for marketers to consider when selecting influencers with whom to work [7]. When an influencer has a large following, their content will be widely circulated [8]. In addition, the number of followers is also considered to be a measure of influencer popularity [1]. To sustain their influence, social media influencers are committed to finding new strategies to grow and maintain their following [9]. It would be rather worthless to work with influencers who

have no influence [10]. Therefore, the size of an influencer's social media following is directly related to their popularity and the efficiency of influencer marketing [11].

Influencer marketing's growing prominence has resulted in an increase in research on this topic. Some studies have examined influencers' source characteristics, such as expertise and credibility [3,12]. Others have investigated the psychological-related influential factors, including parasocial relationships and wishful identification to discover their role in influencer marketing [9]. Numerous studies have been performed to determine the effects of sponsorship disclosure on consumer attitude and behavior [13,14]. The majority of studies have examined influencer marketing as a commercial strategy and explored its effectiveness in the marketing field [10,15]. However, the study on the initial stages of influencers' growth is scarce [8]. This should be investigated further, as the influencers' relationship with their followers is crucial to the development of influencer marketing [11].

Emotional attachment has been identified as a critical factor in a social media influencer's ability to attract and retain followers [16]. Emotions can enable customers to form an attachment to a person or item [17]. Influencers express their personal views on issues via social media and interact with people who see the content they generate through comments and other ways. This gives users a sense of psychological closeness with influencers and consequently builds emotional connections with influencers [18]. Establishing emotional attachment is conducive to enhancing the persuasiveness of influencers to their followers [19].

Information quality can affect the consumers' attitudes, information adoption, and behavioral intentions [20]. The content generated by social media influencers who are dubbed leaders has a considerable impact on users' attitudes and behaviors. Due to the inaccuracy, incompleteness, and unreality of low-quality information, it will have a detrimental effect on customers' perceptions and behavioral decisions, and may even result in dangerous conclusions [21]. High-quality information presents complete, true, timely, and effective content as much as possible, which can entirely meet users' needs and has a positive effect on their behavior [22].To attract users, it is helpful when influencers publish content that users find valuable because they often base their decision-making on the content generated by influencers [8]. However, users are finding it increasingly difficult to locate the information they seek because of information overload [23]. Moreover, the same content shared on social media will have a unique effect on different users [24]. It is interesting to determine what characteristics of influencer-generated content will draw users' attention and affect their behavior, and according to previous studies, it is critical for influencers and their followers to build a strong attachment relationship to achieve effective marketing outcomes [16,25]. As such, the purpose of this study is to investigate the effect of influencer-generated content characteristics on the information quality and consumer emotional attachment based on the concept of information relevance, and consequently to explore the relationship between influencer-generated content and influencer popularity. This study mainly addresses the following three questions: First, how does influencer-generated content affect the emotional attachment that social media users have to influencers? Second, how do the characteristics of influencer-generated content affect the information quality? Third, what are the effects of the emotional attachment and the information quality on influencers' popularity?

This study provides significant theoretical and practical contributions by exploring the important effect of influencer-generated content in enhancing the relationship between influencers and their followers based on the information relevance theory. On a theoretical level, the conceptual model developed in this study explains that the positive impact of influencer-generated content on consumer cognition and emotional attachment eventually increases the influencer's popularity. This demonstrates the critical role of influencer-generated content in the interaction between influencers and their followers. In practice, the research findings provide insights that can assist influencers in optimizing their content and attracting broader public support, so promoting the sustained development of influencer marketing.

## 2. Literature Reviews

### 2.1. The popularity of Social Media Influencers

Social media influencers are defined as individuals who are able to keep in touch with social media users and promote commodities to the targeted consumers. Influencers educate and advise their followers by constantly generating valuable content on social media while trying to develop intimate, long-term relationships with them [2]. Followers are people who voluntarily choose to receive content from the influencer; they are willing to interact with the influencer and other followers who are interested in the same influencer [26,27]. However, influencers can affect their followers' opinions and behaviors by the content they generate [7]. An influencer's relationship with others is the foundation to form influencer marketing [15].

Influencer marketing is a strategy in which brands pay influencers to advertise their commodities on social media in exchange for brand advantages, and it has been shown to be an efficient way for brands to reach customers [1]. According to previous studies, marketers value the number of followers that influencers have when selecting social media influencers to work with [7]. Marketers tend to select influencers who have a large following to promote their brands and products [2]. This is because influencers are regarded as more popular when they have a large following, which can translate into credibility, opinion leaders, parasocial interactions, and more sales, all of which contribute to improved marketing results [26,27]. An influencer with many followers can be regarded as an advertiser of great influence; thus, the follower count is considered an important index for marketers to identify how influential an influencer is [9].

For influencers, the number of followers is crucial because the more followers they have, the more widely their publications are distributed, and the more influential they are [8]. However, influencers' relationships with their followers are more equitable and more unstable than the traditional celebrities' interactions with their fans [9]. Social media users can follow or unfollow a social media influencer at their own discretion, which can affect the size of the influencer's followers [28]. Influencers strive to attract and retain followers, as this is the basis of their influence [10].

In conclusion, an influencer's following reflects both their popularity and their influence [1]. This study examines the influencer's popularity through two variables: consumers' intention to follow or continue following the influencer and consumers' intention to recommend the influencer to others. The increase in followers is a result of social media users following or recommending influencers to others [7,8].

### 2.2. Emotional Attachment

Emotional attachment is a psychological term that refers to a strong emotional connection formed between a person and a particular person or object [29]. Generally, people tend to develop emotional bonds with others who are close to them [27]. Emotional attachment was first studied in the parent-child relationship [28]. However, people's consumption behaviors have been found to exhibit emotional attachment. According to previous research, people can become emotionally attached to brands [30], online communities [30], companies [31], places [32], celebrities [18], and so on. Moreover, companies are increasingly seeking strategies to build strong emotional connections with their consumers, as emotionally involved individuals display stronger brand loyalty, which benefits the company [33]. According to attachment theory, an individual will develop an emotional attachment to an object if the object is able to meet the individual's needs [34].

### 2.3. Information Quality

In social media, information quality can be defined as the perception of the accuracy, consistency, and adequacy of the contents generated by users on social media [35]. This study explains information quality as the extent to which users believe the content generated by social media influencers on their personal pages is accurate, consistent, and adequate. Information quality is seen as a vital aspect in affecting the success of an infor-

mation system and user satisfaction because it can determine the persuasiveness of a piece of information [36,37].

There is plenty of information available online. Getting information is one of the primary reasons individuals utilize social media [38]. Consumers often treat influencers as opinion leaders and refer to the content generated by influencers to help them to make the right decision. Consumers may find the information useful when they perceive high-quality content [39]. High-quality online information can not only be trusted by consumers but also be used to determine whether consumers accept or reject it [40,41].

### 2.4. Influencer-Generated Content

Influencers attract attention and followers by providing content on social media, and they attempt to develop close and long-term relationships with others [1]. According to the previous research, the features of content online might affect consumer perceptions and evaluations [16]. For example, the richness (images and videos) of content and the length of the message might boost post engagement and popularity [39]. Perceived content characteristics (quantity, quality, originality, and uniqueness) can affect influencer-follower relationships [8]. The vividness of brand posts can contribute to the sharing of brand posts [42]. Evidently, content is an important factor that affects users' preferences and behaviors [25].

However, the issue of how influencer-generated content should align with consumer requirements has not been thoroughly explored. The role of influencer-generated content in influencer marketing requires further examination. Additionally, there is presently an inadequate amount of information about how influencers and social media users build strong connections in order to develop and maintain influencer popularity.

To address these research gaps, we propose an empirical research model based on information relevance theory and utilizing the underlying constructs identified from previous research to explain the relationship between influencer-generated content and influencer popularity.

## 3. Theoretical Background and Hypothesis Development

### 3.1. Theory of Information Relevance

Information relevance can be interpreted as the utility of a piece of information as perceived by users [23]. With the increase in online information, it may become difficult for consumers to make decisions, as they have to expend more effort to process the information [43]. Users could not read every message, and they are reluctant to read irrelevant information. Therefore, they may judge the usefulness of information by some particular cues to encourage them to read further [44]. Information relevance has been used to determine if the information meets consumer needs since the 1970s. Previous studies have examined that information relevance can be judged from its topicality, novelty, reliability, and understandability [23]. As enjoyment is the main reason for users to utilize social media, the pleasure of reading information can be employed as another aspect for assessing information relevance [45]. Combining the findings of previous studies and the background of this study, we use the interestingness of content as an aspect of judging the relevance of information.

### 3.1.1. Topicality

Topicality measures the extent to which users perceive information to be pertinent to the topic of their current interest [23]. If individuals accept that the information is related to their topic of interest or the information is highly related to their needs, the information is regarded as topically relevant. As topicality focuses on users' current needs, there is a greater emphasis on timeliness [23]. Topicality can be assumed when searching for information on social media [24]. Moreover, influencers are active in various fields, and they generate content mainly based on their own expertise. Therefore, topicality is not considered in this study.

### 3.1.2. Interestingness

Interestingness of content can be seen as the attraction people feel when reading the content posted on social media; it is the perceived enjoyment, pleasure, and entertainment derived from the content [24]. Researchers discovered that one of the principal factors individuals utilize social media is for enjoyment, and interesting content can satisfy their entertainment needs [45]. Accordingly, sharing interesting information on social media is a useful tactic for attracting users' attention [46]. Previous studies have shown that interesting content is beneficial to content marketing to achieve good results [47].

People are accustomed to using social media for recreational purposes [47]. Marketers can use methods that offer interesting content on social media to help brands to build emotional connections with consumers. This is because interesting information is easier to accept by consumers, meets consumers' entertainment needs, and triggers positive emotional states [48]. Previous research has found that conducting interesting activities on social media can make users pleased, thereby contributing to the development of a strong preference or sentiment for a brand and eliciting an emotional attachment between users and the brand [49]. Based on our knowledge of the existing literature, this study assesses the impact of the interestingness of content generated by influencers on emotional attachment as follows:

**Hypothesis 1 (H1):** *Perceived interestingness of content is positively correlated with the consumers' emotional attachment to influencers.*

### 3.1.3. Novelty

Novelty can be explained as the degree to which a user considers the information as novel or distinct from existing knowledge [23]. In the psychological literature, novelty-seeking is considered an intrinsic motivation for humans [50]. It is probably easier to draw consumers' attention when publishing novel content on social media because they are always attracted to unique and unusual information [51].

If the influencer-generated content is something that people are already familiar with, it may not cause any cognitive change and will reduce people's motivation and interest in disseminating and receiving it [23,24]. People are naturally attracted to unique information when they communicate and are always looking for new stimuli, representing a type of attachment based on the "need for stimulation." The previous study examined that providing users with creative, memorable brand experiences increases brand attachment [52]. Influencers communicate with users primarily through the content they generate. If the influencer generates content that the user has never seen or has seen only infrequently, reading the content provides a novel experience for the user. As a consequence, the influencer may then attract the user's attention and get their approval. Based on the above, we propose:

**Hypothesis 2 (H2):** *Perceived novelty of content positively impacts consumers' emotional attachment to influencers.*

### 3.1.4. Reliability

To remain consistent with the study's context, we focus on the reliability of the content rather than that of the influencer. Reliability is explained as the extent to which information is considered true, accurate, or credible [23]. Influencers commonly generate content based on their own experiences or expertise. If readers believe that the content generated by an influencer is authentic and credible, they will agree with it and pay more attention to the content or influencer. Otherwise, they will not read the content further [53].

Reliability refers to the trustworthiness of information in this study. According to existing studies, influencer-generated content might impact influencers to be seen as opinion leaders by their followers [8]. Reliable information is more compelling and trustworthy

than unreliable information because it can alleviate consumers' perceived confusion about the information and improve its effectiveness [54,55]. Previous research indicated that information reliability, accuracy, and consistency all contribute to keeping information integrity and assists in increasing information quality [56]. Thus, the reliability of information is considered a critical aspect in determining whether or not a user accepts advertising [24]. Previous studies have indicated that consumers are prone to form an emotional attachment to a trusted product or brand [57,58]. Accordingly, this study hypothesizes the following:

**Hypothesis 3-1 (H3-1):** *Perceived reliability of content is positively correlated with consumers' emotional attachment to influencers.*

**Hypothesis 3-2 (H3-2):** *Perceived reliability of content is positively correlated with information quality.*

### 3.1.5. Understandability

Understandability can be defined as the degree to which people believe that information is straightforward to read and understand [23]. Users prefer easy-to-understand information to difficult-to-understand information, while difficult-to-understand information commonly confuses them [59]. As a result, users may reject difficult-to-understand content published on social media [53,60].

Information is a dynamic entity that acquires utility through interpretation by the recipient [61]. Due to the volume of online information, easy-to-understand information can assist users in quickly comprehending and processing the information for their own needs [62]. According to previous studies, content understandability contributes to the quality of information and has a beneficial effect on user adoption of influencer-generated content [63,64]. Additionally, easy-to-understand information can effectively improve users' willingness to read, and positive emotions, which can help strengthen the emotional connection between users and influencers, since influencer-generated content is a source of information for consumers seeking enjoyment and knowledge [24]. In comparison to difficult-to-understand information, easy-to-understand information is easier for readers to comprehend and accept, and it can also improve their reading experience [59]. As a consequence, we proposed the following hypotheses:

**Hypothesis 4-1 (H4-1):** *Content understandability positively affects the consumers' emotional attachment to influencers.*

**Hypothesis 4-2 (H4-2):** *Content understandability has a positive impact on information quality.*

### 3.2. Information Quality

Users are affected by the information quality when searching for information online [65]. People's internal emotional responses can be affected by their judgments of information quality [47]. High-quality information is beneficial for users to better understand the content and provide positive emotional responses, which increases their confidence in making the right decision [22]. Users might increase their interaction with influencers if they believe that influencer-generated content can meet their own needs [66]. Meanwhile, researchers have found that information quality affects users' emotional attachment to influencers [67].

Social media influencers can build their reputation and attract users by generating high-quality content that meets users' needs, as users want to extract useful information from influencer-generated content in order to satisfy their knowledge or entertainment needs and make the best choices [46,68]. The higher the quality of information, the more likely users will trust it. Previous research has indicated that providing users with high-quality information can increase their trust, satisfaction, and intention to continue purchasing [22,62]. That is, influencers can win the trust of users and encourage them to engage in more

positive behaviors by providing high-quality content. Accordingly, this study hypothesizes the following:

**Hypothesis 5-1 (H5-1):** *Information quality has a beneficial effect on consumers' emotional attachment to influencers.*

**Hypothesis 5-2 (H5-2):** *Information quality is positively correlated with the intention to follow/continue to follow an influencer.*

**Hypothesis 5-3 (H5-3):** *Information quality is positively correlated with the intention to recommend the influencer.*

### 3.3. Emotional Attachment

Emotions have a tremendous impact on people's behavior [29]. Social media increases the possibility of reaching influencers and contributes to the building of emotional attachments between influencers and their followers [18]. Users tend to have a positive attitude toward an influencer when they develop an emotional attachment to the influencer [53]. Previous studies have shown that users' emotional attachment to the influencer can positively affect users' purchase intentions, recommendation intentions, and influencer popularity [18]. This is because emotional attachment plays an important role in persuading influencers to their followers [69]. Accordingly, the hypotheses are as follows:

**Hypothesis 6-1 (H6-1):** *Emotional attachment positively affects users' intention to follow/continue to follow the influencer.*

**Hypothesis 6-2 (H6-2):** *Emotional attachment is positively correlated with the intention to recommend the influencer.*

### 3.4. Research Model

We developed the research model (Figure 1) for this study based on the theoretical comprehension of the pertinent literature. The term "information quality" measures the accuracy of content generated on the personal accounts of social media influencers in the study. However, the information interestingness and information novelty have a minimal impact on information accuracy. As a result, we put greater emphasis on the impact of information reliability and understandability on information quality in this study, rather than the impact of information interestingness and information novelty on information quality.

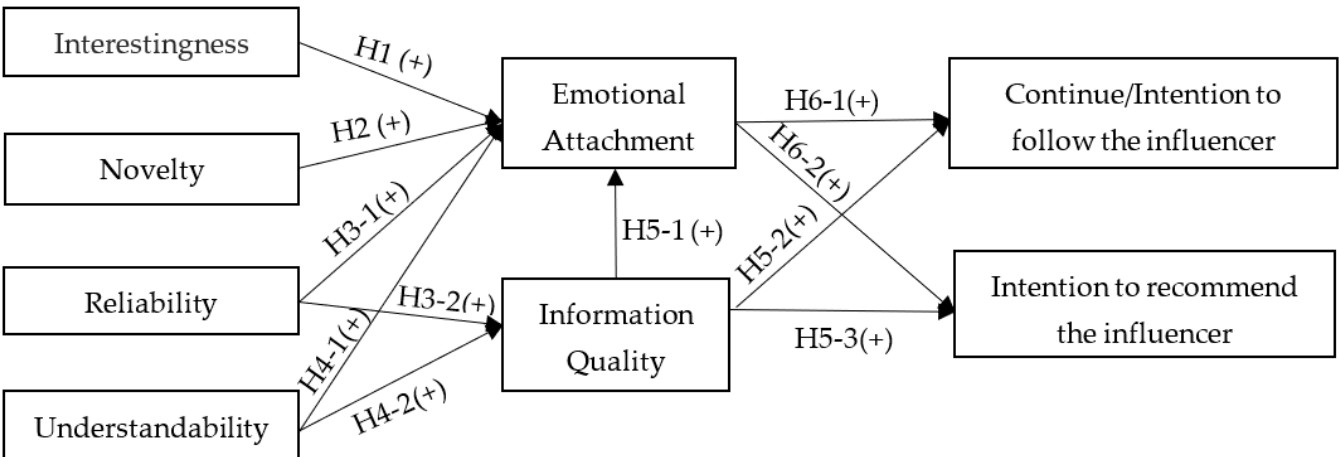

**Figure 1.** Research Model.

## 4. Methodology

### 4.1. Measures

The measurement instruments were taken from previous studies and tailored to satisfy the requirements of this study. The scales developed by Chen and Lin (2018) [47], and Zhao and Zhang (2020) [70] served as the foundation for developing a measurement scale for the concept of content interestingness, while the content novelty and the content reliability were determined using the scale derived from Xu and Chen (2006) [23], and the content understandability scale items were adopted from Filieri and McLeay (2014) [63]. The information quality was measured using the adapted scales of Song et al., (2017) [68], while the scale adapted from Sánchez-Fernández and Jiménez-Castillo (2021) [18] was used to measure emotional attachment. Finally, continue/intention to follow the influencer was examined using the scales from Belanche (2021) [10], and intention to recommend the influencer was evaluated by using the scale adapted from Song et al., (2017) [68].

### 4.2. Sample, Data Collection, and Validation Method

The data for this study were collected in Korea. First, we translated the measurement items into Korean to guarantee the smooth performance of the survey. Thereafter, a Korean scholar who spoke English well was required to review the Korean version to ensure the accuracy of the translation. We optimized the expression according to suggestions from a professor in business administration. Finally, we performed a pilot test before the official distribution of the questionnaires, with three students who followed social media influencers, invited to attend and tasked with determining the readability of each item. The phrasing of the measures was further changed in light of the test results.

We used Google Forms to create a three-part online questionnaire. In the first part, we defined the term "social media influencer" and gave participants two examples to help them better understand our research background. Thereafter, we required the participants to name their favorite social media influencers. We filtered out responses that incorrectly named celebrities rather than influencers to protect the quality of our data. The second part requested participants to respond to questions about the measurement items that rely on the influencer they named first. The third part included demographic questions about the participants. With the exception of the individuals' demographic variables, all items were assessed using a 5-item Likert scale ranging from 1 (strongly disagree) to 5 (strongly agree).

Participants were recruited using the snowball sampling method. Initially, the researchers recruited participants by sharing questionnaire links and publicizing the study's purpose on their own social media accounts. Second, participants were invited to share questionnaire links with friends and relatives to increase the number of participants and eliminate selection bias. Additionally, participation was anonymous and voluntary in order to avoid bias. A total of 368 responses were obtained. The remaining 280 valid questionnaires were analyzed after filtering. The participants were 61% female, and 59.3% of the participants were aged 20–29, while the next group of participants was aged 30–39 (22.9%). Instagram was the preferred platform for respondents (53.2%) to follow influencers, followed by YouTube (33.6%) and Facebook (13.2%). According to Info Cubic, Instagram, YouTube, and Facebook are the most commonly used social networking platforms in Korea. As far as influencer marketing is concerned, Instagram is by far the most extensively utilized and significant social media channel [6]. Forty percent of the participants spend 30 min to 1 h per day reading influencer-generated content. The participants focus on a wide range of areas, with live streaming (85.4%) and product reviews (84.3%) accounting for the largest share. Table 1 summarizes the participants' demographic information.

**Table 1.** Participant Demographics.

| Measure | Item | Count (n = 280) | Percentage (%) |
|---|---|---|---|
| Gender | male | 110 | 39% |
| | female | 170 | 61% |
| Education | High school | 33 | 11.80% |
| | Undergraduate | 185 | 66.10% |
| | Postgraduate | 62 | 22.10% |
| Age | 10–19 years old | 39 | 13.90% |
| | 20–29 years old | 166 | 59.30% |
| | 30–39 years old | 64 | 22.90% |
| | 40–49 years old | 11 | 4% |
| Which social media platform do you use the most to follow influencers? | YouTube | 94 | 33.60% |
| | Facebook | 37 | 13.20% |
| | Instagram | 149 | 53.20% |
| How much time do you spend on influencers' posts per day? | <30 min | 78 | 27.90% |
| | 30 min–1 h | 112 | 40.00% |
| | >1 h | 90 | 32.10% |
| Please select the category of content that you are primarily interested in. | Music | 96 | 34.30% |
| | Sports | 173 | 62% |
| | Game | 168 | 60.00% |
| | Film | 115 | 41.10% |
| | Fitness | 203 | 73% |
| | News | 171 | 61.10% |
| | Product reviews | 236 | 84.30% |
| | Live streaming | 239 | 85.40% |
| | Study | 160 | 57.10% |
| | Vicarious experience video (travel, food, etc.) | 146 | 52.10% |

Furthermore, we correlated participants' responses to influencer categories to acquire a deeper understanding of the survey results. Influencers can be classified into the following categories: lifestyle, fashion, beauty, sports/fitness, travel, parenting, photography, music, food, gaming, pet, virtual, and real estate [71]. According to the results, influencers in the food (28.6 %), travel (23.5 %), and sport/fitness (22.8 %) categories accounted for a larger percentage. This reflects that the participants place a higher premium on life quality. Additionally, this could be due to the fact that the participants aged 20–29 accounted for a sizable portion of the survey.

We selected PLS-SEM (Partial Least Squares Structural Equation Modeling) to assess the research model and its corresponding hypotheses. PLS-SEM can be used to confirm the construct validity of an instrument and to analyze the structural relationships between constructs; it is also an effective method for analyzing composite-based path models [72,73]. Not only does PLS-SEM have high statistical power for research with small sample sizes, but it is also widely regarded as the preferred tool for studies with the aim of theory development and exploration [73]. Considering the exploratory nature of this study, PLS-SEM is thus more suitable for model estimating in our study. SmartPLS was used as the software to perform two-stage data analysis models based on PLS-SEM (i.e., the measurement model and structural model).

## 5. Results

### 5.1. Measurement Model Assessment

To examine the measurement model, we first determined the item loadings on the relevant constructs in order to identify the indicator's reliability. Second, we assessed each construct's internal consistency using composite reliability and Cronbach's alpha. Third, we examined the convergent validity by calculating the average variance extraction (AVE)

values for each construct. Fourth, the heterotrait-monotrait (HTMT) ratio of the correlations was used to assess discriminant validity.

As shown in Table 2, all item loadings are more than 0.7, confirming the reliability of items. The values of composite reliability varied from 0.834 to 0.914, and Cronbach's alpha (α) for each construct was larger than the suggested threshold of 0.7. These assessments confirmed the internal consistency of the measures for each construct. Additionally, all the AVE values were significantly above the needed minimum threshold of 0.5, showing that all measures have a high degree of convergent validity. Table 3 summarizes the findings of the discriminant validity analysis using the HTMT criterion. According to the results, each value is less than the critical value of 0.85 [74]. Additionally, we conducted HTMT inference using bootstrapping, and all confidence interval values are less than 1, indicating that discriminant validity is established using HTMT as well.

**Table 2.** Results of measurement model analysis.

| Factor | Item | Factor Loading | VIF | Cronbach's Alpha (α) | CR | AVE |
|---|---|---|---|---|---|---|
| Interestingness (Int) | Int1 Influencer-generated content is interesting. | 0.708 | 1.42 | 0.780 | 0.871 | 0.692 |
| | Int2 Influencer-generated content is attractive. | 0.836 | 1.802 | | | |
| | Int3 I like the influencer-generated content. | 0.822 | 1.847 | | | |
| Novelty (Nov) | Nov1 The influencer-generated content is unique. | 0.706 | 1.578 | 0.809 | 0.872 | 0.630 |
| | Nov2 There is a lot of new information in the influencer-generated content. | 0.748 | 1.505 | | | |
| | Nov3 Influencer-generated content has a great deal of information that I was previously unaware of. | 0.831 | 1.973 | | | |
| | Nov4 Influencer-generated content satisfies my sense of curiosity. | 0.773 | 1.791 | | | |
| Reliability (Re) | Re1 I think the influencer-generated content is accurate. | 0.813 | 2.067 | 0.841 | 0.903 | 0.757 |
| | Re2 I think the influencer-generated content is consistent with facts. | 0.828 | 2.219 | | | |
| | Re3 I think the influencer-generated content is reliable. | 0.783 | 1.816 | | | |
| Understandability (Und) | Und1 The influencer-generated content is easy to understand. | 0.784 | 2.003 | 0.846 | 0.905 | 0.760 |
| | Und2 The influencer-generated content is easy to interpret. | 0.831 | 2.114 | | | |
| | Und3 The influencer-generated content is easy to read. | 0.734 | 2.006 | | | |
| Emotional attachment (Emo) | Emo1 I feel emotionally connected to the influencer. | 0.713 | 1.545 | 0.823 | 0.881 | 0.650 |
| | Emo2 I am very attached to the influencer. | 0.834 | 1.84 | | | |
| | Emo3 The influencer is special for me. | 0.784 | 1.801 | | | |
| | Emo4 I miss the influencer if they don't post or if I can't see their postings. | 0.794 | 1.77 | | | |
| Information quality (Inf) | Inf1 I am satisfied with the information quality of influencer-generated content. | 0.716 | 1.302 | 0.704 | 0.834 | 0.626 |
| | Inf2 Influencer-generated content could exactly report what I need. | 0.795 | 1.424 | | | |
| | Inf3 Influencer-generated content could provide precise information I need. | 0.745 | 1.418 | | | |
| Continue/Intention to follow the influencer (Con) | Con1 I intend to continue following this influencer in the near future. | 0.812 | 1.845 | 0.825 | 0.894 | 0.739 |
| | Con2 I predict that I will continue following this influencer. | 0.871 | 2.201 | | | |
| | Con3 I am likely to look for new content published by this influencer. | 0.781 | 1.758 | | | |
| Intention to recommend the influencer (Inten) | Inten1 I would refer this influencer to others. | 0.820 | 2.32 | 0.859 | 0.914 | 0.780 |
| | Inten2 I would say positive things about this influencer to other people. | 0.754 | 2.134 | | | |
| | Inten3 This influencer is someone I would suggest to others. | 0.807 | 2.085 | | | |

**Table 3.** Assessment of discriminant validity using HTMT.

|  | Con | Emo | Inf | Int | Inten | Nov | Re | Und |
|---|---|---|---|---|---|---|---|---|
| Con |  |  |  |  |  |  |  |  |
| Emo | 0.157 |  |  |  |  |  |  |  |
| Inf | 0.464 | 0.426 |  |  |  |  |  |  |
| Int | 0.309 | 0.229 | 0.298 |  |  |  |  |  |
| Inten | 0.447 | 0.413 | 0.337 | 0.363 |  |  |  |  |
| Nov | 0.254 | 0.217 | 0.236 | 0.461 | 0.362 |  |  |  |
| Re | 0.257 | 0.325 | 0.267 | 0.367 | 0.502 | 0.423 |  |  |
| Und | 0.36 | 0.36 | 0.299 | 0.426 | 0.627 | 0.379 | 0.601 |  |

Note: HTMT = heterotrait-monotrait criterion. Int—Interestingness; Nov—Novelty; Re—Reliability; Und—Understandability; Emo—Emotional attachment; Inf—Information quality; Con—Continue/Intention to follow the influencer; Inten—Intention to recommend the influencer.

### 5.2. Structural Model Assessment

Collinearity must be examined prior to examining structural relationships to ensure that it does not bias the regression findings [73]. This may be determined by the use of the variance inflation factor (VIF). As demonstrated in Table 2, all VIF values in the research model's predictor structure were less than the threshold of 5, indicating no serious multicollinearity issues.

To determine the explanatory power and predictive accuracy of the model, the squared multiple correlation ($R^2$) and Stone–Geisser's $Q^2$ values [75–77] were calculated using the bootstrapping and blindfolding procedures in SmartPLS. Table 4 presents the $R^2$, $R^2$ adjusted, and $Q^2$ values for the endogenous constructs. $R^2$ measures the proportion of variance explained by the model's independent constructs. According to the findings, content characteristics (interestingness, novelty, reliability, and understandability) predicted 18% of the variance in emotional attachment. Content reliability and understandability explained around 7% of the variance in information quality. Emotional attachment and information quality explained 13% of the variance in intentions to follow an influencer and 15% of the variance in intention to recommend the influencer to others. The $Q^2$ value should be larger than 0 to show the prediction accuracy of the developed structural model [78]. As a result, our study model exhibits both explanatory power and predictive relevance for endogenous constructs.

**Table 4.** $R^2$, $R^2$ Adjusted, and $Q^2$.

|  | $R^2$ | $R^2$ Adjusted | $Q^2$ |
|---|---|---|---|
| Emotional Attachment | 0.185 | 0.17 | 0.108 |
| Information Quality | 0.073 | 0.066 | 0.038 |
| Intention to recommend the influencer | 0.15 | 0.144 | 0.113 |
| Continue/Intention to follow the influencer | 0.132 | 0.125 | 0.087 |

A bootstrapping approach in SmartPLS with 5000 subsamples was used to determine the significance of the path coefficients in order to identify the structural validity of the model. The results are presented in Figure 2. A Table 5 summarizes the causal relationships between the constructs and the hypothesis test. As shown in Figure 2, the path coefficient for H1 is considerable and positive ($\beta = 0.019$, $p < 0.05$), which indicates that information interestingness has a significant effect on users' emotional attachment to influencers, thereby supporting H1. Information novelty ($\beta = 0.039$, $p < 0.001$) can significantly affect users' emotional attachment to influencers; thus, H2 is supported. The findings confirm that a positive and significant association exists between information reliability and emotional

attachment to influencers ($\beta$ = 0.108, *p* < 0.001); supporting H3-1. The hypothesis that information understandability increases users' emotional attachment to influencers (H4-1) is also supported, with a significant path coefficient ($\beta$ = 0.188, *p* < 0.001). Moreover, the path coefficient ($\beta$ = 0.252, *p* < 0.001) between information quality and users' emotional attachment to influencers is substantial; thus, H5-1 is supported. Therefore, we conclude that information relevance (interestingness, novelty, reliability, and understandability) might have a major impact on users' emotional attachment to influencers, and information quality can increase users' emotional attachment to influencers.

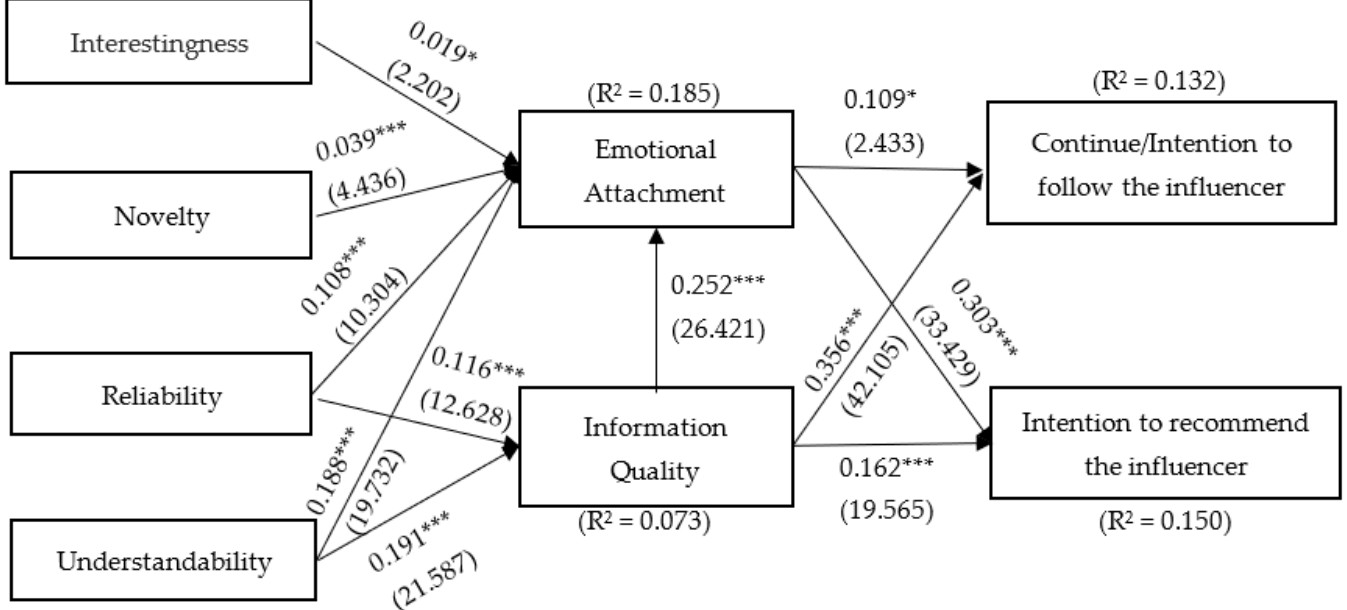

**Figure 2.** Path coefficient. Note: *: *p* < 0.05, ***: *p* < 0.001. The numbers in the parenthesis mean t-value.

**Table 5.** Structural relationships and hypothesis testing.

| Hypothesis | Relationship | Path Coefficient | t-Value | Sig. |
|---|---|---|---|---|
| H1 | Interestingness → Emotional attachment | 0.019 | 2.202 | Supported |
| H2 | Novelty → Emotional attachment | 0.039 | 4.436 | Supported |
| H3-1 | Reliability → Emotional attachment | 0.108 | 10.304 | Supported |
| H3-2 | Reliability → Information quality | 0.116 | 12.628 | Supported |
| H4-1 | Understandability → Emotional attachment | 0.188 | 19.732 | Supported |
| H4-2 | Understandability → Information quality | 0.191 | 21.587 | Supported |
| H5 | Information quality → Emotional attachment | 0.252 | 26.421 | Supported |
| H5-1 | Information quality → Continue/Intention to follow the influencer | 0.356 | 42.105 | Supported |
| H5-2 | Information quality → Intention to recommend the influencer | 0.162 | 19.565 | Supported |
| H6-1 | Emotional attachment → Continue/Intention to follow the influencer | 0.109 | 2.433 | Supported |
| H6-2 | Emotional attachment → Intention to recommend the influencer | 0.303 | 33.429 | Supported |

In addition, there is a significant path coefficient ($\beta$ = 0.116, *p* < 0.001) between information quality and information reliability; thus, H3-2 is supported. Information under-

standability ($\beta$ = 0.191, $p$ < 0.001) might have a considerable impact on information quality; hence, H4-2 is supported. The results show that the understandability of information seems to be a strong predictor of information quality.

Emotional attachment ($\beta$ = 0.109, $p$ < 0.05) and information quality ($\beta$ = 0.356, $p$ < 0.001) can increase users' intention to follow influencers; thus, H6-1 and H5-2 are supported. Finally, emotional attachment ($\beta$ = 0.303, $p$ < 0.001) and information quality ($\beta$ = 0.162, $p$ < 0.001) both have a positive effect on users' intention to recommend influencers, which support H6-2 and H5-3.

## 6. Discussions

This study tested a research model based on the concept of information relevance of the relationship between influencer-generated content and influencer popularity. Influencer popularity is quantifiable in terms of the number of influencer followers, and social media users following or recommending an influencer to others are the primary ways for an influencer's follower count to increase [7,8]. The findings of this study show that influencer-generated content has a considerable effect on influencer popularity since it affects the information quality and emotional attachment, two factors that motivate people to follow or recommend an influencer to others.

The first important finding is that enhancing users' emotional attachment to influencers can increase users' intention to follow or recommend influencers to others, thereby expanding the visibility and follower size of influencers. And compared with information quality, emotional attachment between users and influencers is more likely to inspire users to recommend influencers to others. The easily accessible nature of social media enables people to follow influencers at any time and from any location, which may help eliminate psychological distance and foster emotional relationships between users and influencers. Influencers can more effectively persuade social media users to respond positively if there is a stronger emotional attachment between social media influencers and users [16]. These results indicate that user emotion has an impact on influencer popularity.

The second important finding is that the quality of influencer-generated content has a considerable effect on users' emotional attachment to an influencer and on their decision to follow or recommend an influencer. Furthermore, information quality is more likely to drive people to follow an influencer than emotional attachment is. According to previous research, people refer to social media influencers as opinion leaders because they anticipate that the influencer will provide them with important information to assist them in making wise choices. As a result, offering high-quality and valuable information to users is more likely to assist influencers in obtaining user trust and support, as well as developing a stronger emotional attachment with users [79,80]. These results indicate that social media users expect to obtain beneficial information from influencers and that user cognitions have a major impact on influencer popularity.

Regarding the third crucial finding, previous research explored the attributes of information relevance from a utilitarian perspective [23]. By combining our findings with those from previous research, we found that interestingness should be considered as a component of information relevance in the context of influencer marketing. Also, it is essential to explore other dimensions of information relevance in light of the particularity of the social media environment. Additionally, we discovered that while interestingness, novelty, reliability, and understandability all contributed to emotional attachment and information quality in this study, understandability had the greatest effect on emotional attachment and information quality. That could be because people tend to seek out useful information quickly and easy-to-understand information can assist users to conserve their energy and time in today's fast-paced and information-overloaded environment [64]. These findings can assist influencers in generating content.

Fourth, previous research suggested that $R^2$, which is used to evaluate prediction accuracy, ranges from 0 to 1, with higher values indicating better prediction performance [78]. The $R^2$ values in this study are not exceptionally high as all the values of $R^2$ are less than 0.2.

Although this finding is applicable to the scope of influencer marketing research, the results in other fields of research may be different because the value of $R^2$ is usually correlated to the complexity of the model and the research discipline [78].

### 6.1. Theoretical Implications

First, this study analyzed the effect of influencer-generated content on influencer popularity and improves the understanding of recent research trends in influencer marketing. Many of the existing studies on influencer marketing regard influencer marketing as a marketing strategy and examine its impact on brands and products [10,15]. In fact, influencer-generated content serves as the basis for marketing [16]. The findings of this study shed new light on the relationship between influencer-generated content and influencer popularity and expand the influencer-marketing literature.

Second, this study uses information relevance theory to investigate the features of influencer-generated content from the perspective of satisfying consumers' information needs. The results show that these characteristics can affect consumers' emotions and subsequently affect influencer popularity. Previous studies have examined the features of influencer-generated content, such as originality, creativity, and content quantity [8,39]. The findings may aid in the comprehension of influencer-generated content.

### 6.2. Managerial Implications

The findings have practical implications for influencers and businesses seeking to collaborate with influencers.

First, the survey results indicate that influencer-generated content can help influencers develop emotional connections with users when it meets consumers' information needs. Understandability has the greatest impact on emotional attachment among the four factors (i.e., interestingness, novelty, reliability, and understandability) examined in this study. Therefore, influencer-generated content should be easy for users to understand, especially in fields with strong expertise that users are not usually exposed to. The reliability of information also has a greater impact on emotional attachment and information quality. Accordingly, influencers can explain some evidence when they post content to increase the reliability of the content and the trust of users. Meanwhile, influencers should pay attention to whether the content is new and interesting because if the influencer-generated content is neither innovative nor well-known by users, it will be rejected. In addition, these results show that influencers must carefully manage their emotional connections with their followers and the information quality to maintain their influence.

Second, for brands, marketers can check whether influencer-generated content can attract consumers' attention when selecting influencers to work with, particularly from the perspective of users' information needs. Because influencers receive information from marketers and then share that information with other users [81], they can play a critical part in information propaganda for the brand. Marketers must determine whether their use of influencers in digital marketing strategies is effective at eliciting positive consumer reactions to the brand.

### 6.3. Limitations and Future Research Directions

First, we apply the concept of information relevance to examine influencer-generated content across all categories of influencers. Future research could investigate whether the model proposed in this study is applicable to content generated by various types of influencers, as there are numerous types of social media influencers (e.g., beauty influencers, entertainment influencers, etc.). Second, this study looks into the influencers' relationship with social media users from the standpoint of users' information needs. Future research can incorporate moderating variables, such as user motivation (hedonic motivation/utilitarian motivation), into the connection between content characteristics and influencer popularity in order to examine the relationship between influencers and others in greater detail. Third, this study evaluated the effect of influencer-generated content on

emotional attachment. Future research can explore whether influencer-generated content affects consumers' attitudes toward influencers to deepen the understanding of influencer-generated content. Fourth, because the participants in this study were Korean, the results may vary in other countries due to differing cultural values.

Although this study is not without a few limitations, we expect that it will contribute to research in this important field and that the findings of this study will be beneficial as a guideline for future research.

## 7. Concluding Remarks

Influencer popularity has a significant effect on the development of influencers and influencer marketing. This study examined the relationship between influencer-generated content and influencer popularity based on information relevance theory. The findings indicate that influencer-generated content can affect consumer emotions and cognitions, hence affecting influencer popularity. Among them, the understandability of the content has a greater impact on emotional attachment and information quality than the content's interestingness, novelty, and reliability. Thus, social media users anticipate more valuable information from influencers based on the context of influencer marketing. With the increasing development of influencer marketing, further study is necessary to understand the factors that contribute to influencer popularity.

**Author Contributions:** Conceptualization, X.Z. and J.C.; methodology, X.Z. and J.C.; software, X.Z.; formal analysis, X.Z.; investigation, X.Z.; data curation, J.C.; writing—original draft preparation, X.Z.; writing—review and editing, J.C.; supervision, J.C. All authors have read and agreed to the published version of the manuscript.

**Funding:** This work was supported by the Ministry of Education of the Republic of Korea and the National Research Foundation of Korea (NRF-2020S1A5A2A01041510). This research was funded by the Soonchunhyang University.

**Institutional Review Board Statement:** Not applicable.

**Informed Consent Statement:** Not applicable.

**Data Availability Statement:** Not applicable.

**Conflicts of Interest:** There is no declaration of interest.

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
