# Peer review of "The Importance of Social Influencer-Generated Contents for User Cognition and Emotional Attachment: An Information Relevance Perspective"

_sustainability, doi:10.3390/su14116676_

Round 1

Reviewer 1 Report

This study analyzes the relationship between content generated by social media influencers and followers' emotional attachment and perceived quality of information, as well as their intention to follow/recommend the influencer. With the growing influence of social media marketing, the findings presented in this study are of theoretical and practical significance.

However, as an academic study, there are some points that need to be revised. 

First, it is necessary to explicitly present the research gap in the extant literature that this study addresses, and to situate the research questions of this study there.

Next, a more detailed description of the survey target is required, including what categories of social media influencers will be described in this survey. 

It is also necessary to explain why the application of PLS-SEM used in this analysis is appropriate and to report items that are recommended to be reported by PLS-SEM.

Finally, the discussion should explicitly state what the novel findings of this study are.

Below are detailed comments tied to page and line numbers.

--------------------

% page 1, line 2: Title
The capitalization manner should be consistent.  

% page 1, line 25: Abstract
Regarding "This study contributes us...", who is "us"? The pronoun should be explicitly described.

% page 2, line 82: 1. Introduction
The authors described they will examine the relationship between contents of social media influencers and emotional attachment. However, the research questions are about popularity. The authors should structure the existing issues addressed in this study to make them clearer and connect them to the research questions as appropriate.

% page 2, line 87: 1. Introduction
Authors should also describe what implications are expected from the resolution of the research questions in this study. This will clarify the scope of this research.

% page 2, line 88: 2. Theoretical background
Although the authors have described extant literature in detail, authors should not directly link them to the viewpoints addressed in this study, but should first clarify the issues of extant literature, and then structure how this study will address those issues.

% page 3, line 98: 2. Theoretical background
Regarding "...the foundation to form Influencer marketing", capitalization manner should be consistent.  

% page 5, line 242: 3. Research model and hypotheses
Regarding "Perceived novelty is positively impacts consumers' emotional attachment...", is the grammar correct? Sentences should be written with accurate grammatical expressions.

% page 7, line 338: 4.1 Data Collection
Regarding "...indicating strong", is it correct? Isn't that "weak"?

% page 7, line 340: 4.1 Data Collection
The authors wrote that they had the influencer's name described. Although there are many different categories of social media influencers, this study can be used effectively by presenting information on the influencer category that this study is targeting.

% page 8, line 356: 
Before evaluating the specific measurements, authors should first explain what method of analysis will be applied in this study. The methodology they choose will affect how they evaluate the measurement model. And, of course, they should also explain why the method is best suited for application to the current study.

% page 8, line 356: 
In the assessment of the PLS-SEM measurement model, it is first necessary to separate reflective and formative models. Furthermore, for reflective models, please report the HTMT(heterotrait-monotrait) ratio, which is now also recommended. 

Please see the references, e.g.: 
Hair, J. F., Risher, J. J., Sarstedt, M., & Ringle, C. M. (2019). When to use and how to report the results of PLS-SEM. European Business Review, 31(1), 2-24. doi:10.1108/ebr-11-2018-0203
Hair, J. F., Hult, G. T. M., Ringle, C., & Sarstedt, M. (2016). A primer on partial least squares structural equation modeling (PLS-SEM): Sage publications.

% page 11, line 424: 
Significant paths have been found, but R2s are not very high. The difference in the target influencers for the evaluation among the respondents may have influenced the results. Therefore, a more effective use of this study might be made by making comparisons between men and women or by dividing the category of influencers by one more level. Please consider this.

% page 11, line 431: 
What perspective implication is described in this section? Implication is also described in the next concluding section. The structure of this point should be consistent. 

% page 11, line 431: 
In the interpretation of hypothesis testing, the results obtained should not be explained as is, but should be described in such a way that it is clear from what perspective the new findings were obtained.

% page 11, line 442: 
I still didn't quite understand the description of "pupularity". Since the outcomes addressed in this study are the following intention and recommendation intention, they may be described directly at the level of such constructs. Please consider this point as well.

% page 12, line 466:  
Implications are explained in the conclusion, but in some cases this is content that could be included in discussion section. Journal guidelines should be followed, but make sure the structure is consistent.

--------------------

Author Response

Thank you very much for your kindly comments on our manuscript. There is no doubt that these comments are very valuable and very helpful for revising and improving our manuscript. 

Reviewer 2 Report

The topic is very interesting and I enjoyed while reading the paper. But, I have few questions that should be addressed: 

First is that the authors should developed the paper in a way that it is suitable for publication in Sustainability. In current form, the paper is more tilted towards Marketing domain, not towards sustainability. 

Second, the paper should need a comprehensive theory and a dedicated section in literature review section. 

Third, the theoretical and managerial implication sections are missing or required comprehensive discussion. 

Author Response

Thank you very much for your kindly comments on our manuscript. There is no doubt that these comments are very valuable and very helpful for revising and improving our manuscript. Please see the attachment.

Reviewer 3 Report

Dear authors, the research proposed by you seems very interesting and well structured.

I recommend writing the abstract in a single paragraph.

I recommend setting the confidence interval and margin of error for the sample. The limitations of the research should also include the sample size, which is insufficient to be statistically representative.

A final recommendation would be to make a possible comparison with similar or different situations in other geographical regions on the topic discussed.

Author Response

Thank you for your valuable comments. We have worked hard to ensure that this revision meets your expectations. We hope you agree that the revision successfully addresses the concerns expressed by your comments.

Round 2

Reviewer 1 Report

Thank you for revising the manuscript based on the peer review comments. The argument of the manuscript has been clarified. 

However, there are still several points that need to be corrected. Please take the following comments into consideration.

-----------------

# page 7, line 384: 3.2 Information quality/Par.3: 

The way of line break of this paragraph ("Accordingly, this study...") is not standardized with the other subsections. If there is no particular intention, the author(s) should be unified.

# page 7, line 387: 3.2 Information quality/Par.3: 

The rules for numbering hypotheses here are unclear. It should eventually be corrected with a unified numbering manner. 

# page 7, line 407: 3.4 Reserach Model/Par.1: 

The sentence "This study does not consider..." does not sound like a rational reason. A more rational reason for building such a research model should be explained.

# page 7, line 424: 3.4 Reserach Model/Fig.1

Some hypotheses explain the correlation perspective and thus cannot be directly adapted to the causal structure shown in this figure. This is because correlation has no direction. Therefore, if some hypotheses are to be aligned with this research model, they must be assumed to be causal.

# page 7, line 424: 4. Methodology

In the review manuscript of my side, the line does not appear to be broken.

# page 8, line 467: 4.2.Sample.../Par.3: 

The author(s) should explain from what population they sampled the data in Korea this time. Was the data obtained by random sampling from the entire Korean population?

# page 8, line 579: 

The values of R2 do not seem high. This perspective should also be mentioned in the discussion section. For instance, since this is a survey of various types of social media influencers, please consider the possibility that it may be difficult to extract a unified trend.

# page 13, line 612: Fig.2

It should describe what the numbers in the parenthesis of the path represent.

# page 13, line 633: 6. Discussion/Par.2: 

The author(s) claim that information quality has more influence on recommendation intention than emotional attachment, but Figure 2 shows that emotional attachment seems to have a stronger influence on recommendation intention than information quality. The claim here should be stated in a way that clearly shows how it is based on the results. 

-----------------

Reviewer 2 Report

I have reviewed the revisions, and all comments have been resolved.

Author Response

Thank you for your appreciation of our paper. We have benefited greatly from your thoughtful comments on improving it.